# Influence of Different Vegetation Types on Soil Physicochemical Parameters and Fungal Communities

**DOI:** 10.3390/microorganisms10040829

**Published:** 2022-04-16

**Authors:** Xin Sui, Xiannan Zeng, Mengsha Li, Xiaohong Weng, Beat Frey, Libin Yang, Maihe Li

**Affiliations:** 1Engineering Research Center of Agricultural Microbiology Technology, Ministry of Education, Heilongjiang University, Harbin 150080, China; xinsui_cool@126.com (X.S.); xiaohongweng@126.com (X.W.); 2Heilongjiang Provincial Key Laboratory of Ecological Restoration and Resource Utilization for Cold Region, School of Life Sciences, Heilongjiang University, Harbin 150080, China; 3Swiss Federal Research Institute WSL, 8903 Birmensdorf, Switzerland; beat.frey@wsl.ch; 4Institute of Crop Cultivation and Tillage, Heilongjiang Academy of Agricultural Sciences, Harbin 150088, China; zengxiannanzxn@163.com; 5Institute of Nature and Ecology, Heilongjiang Academy of Sciences, Harbin 150040, China; 6Key Laboratory of Geographical Processes and Ecological Security in Changbai Mountains, Ministry of Education, School of Geographical Sciences, Northeast Normal University, Changchun 130024, China; 7Erguna Forest-Steppe Ecotone Research Station, CAS Key Laboratory of Forest Ecology and Management, Institute of Applied Ecology, Chinese Academy of Sciences, Shenyang 110164, China

**Keywords:** cold temperate climate, diversity, forest types, fungal community, restoration

## Abstract

This study assessed the effects of *Betula dahurica* (BD), *Betula platyphylla* (BP), *Larix gmelinii* (LG), *Quercus mongolica* (QM), and a mixed conifer–broadleaf forest composed of LG and QM (LGQM) on the soil physicochemical parameters and community structure of fungi in the Zhongyangzhan Black-billed Capercaillie Nature Reserve. Fungal community structures were characterized via ITS rRNA sequencing. The effects of soil parameters on the community structure of soil fungi were assessed by Pearson correlation analysis and redundancy analysis (RDA). LGQM exhibited lower C/N, available nitrogen (AN), total phosphorus (TP), and available phosphorus (AP) compared with the QM broadleaf forest. The fungal Shannon and Simpson diversity indices were highest in BP, whereas LG exhibited the highest ACE index. The Basidiomycota, Ascomycota, Mortierellomycota, and Mucoromycota fungal phyla were dominant across all vegetation types. Each of the different vegetation types studied herein exhibited a unique fungal community structure. The RDA results indicated that fungal community structures were primarily shaped by the total N, available N, and available P of soil. Our findings thus indicated that forests restored with different species of trees may exhibit variations in soil quality and characteristics despite sharing the same climate. Furthermore, broadleaved and coniferous forests exhibited a unique fungal community diversity and composition.

## 1. Introduction

Forest overexploitation and other human activities have severely damaged the autochthonous vegetation of temperate regions in China, which has negatively impacted biodiversity and ecological functions [1] (Liu et al., 2019). Vegetation restoration is a crucial strategy to preserve ecological functions [2,3,4] (Peng et al., 2013; Nunezmir et al., 2015; Li et al., 2017), biological diversity [5] (Gallardo et al., 2012), and ecosystem services [6] (Oh and Lee, 2009). Vegetation restoration also enhances several soil properties including bulk density [7] (Zhang et al., 2018) and texture and soil/water conditions [8] (Wu et al., 2016), as well as nitrogen, organic carbon, and phosphorous [9] (Xu et al., 2015). Soil microorganism community structure and dynamics are largely shaped by the feedback between plants and soil [10] (Schweitzer et al., 2008). The soil microflora constitutes an important link between plants and soil processes and therefore contributes greatly to the biotransformation of nutrients in soil [11] (Cheng et al., 2013). In particular, nutrient storage is directly determined by the microbial biomass in soils, and microbial metabolism can also indirectly affect nutrient biotransformation [12,13] (Turner et al., 2019; Fan et al., 2020). Nevertheless, despite the critical contributions of soil microorganisms to the global biogeochemical cycles and the success of ecological restoration efforts [14] (Paul et al., 2010), very few studies have characterized the effects of vegetation and soil properties on the structure of soil microbial communities.

Soil fungi are key contributors to several ecological processes [15] (Yue et al., 2021). In particular, fungal communities are intimately linked to a variety of biotic and abiotic factors, such as nutrients [16] (Ding et al., 2021), elevation [17] (Liu et al., 2018), and plant species [18] (Marta et al., 2021). A growing body of evidence has demonstrated that the structure of fungal communities affects several soil properties including nitrogen availability [19] (Frey et al., 2004), texture [20] (Chaudhary et al., 2018), and pH [21] (Kang et al., 2021). Furthermore, vegetation cover also greatly affects soil physicochemical properties. Therefore, vegetation restoration may modify soil properties, leading to changes in the fungal community, which in turn affects restoration success [22] (Wang, 2011). The replanting of native vegetation was recently linked to drastic effects on soil fungal community composition [23] (Yan et al., 2018). Furthermore, a number of studies have characterized the interactions between soil properties and fungal communities in grasslands [24] (Harrison & Bardgett, 2010). Changes in the fungal community composition and diversity during forest vegetation change are key indicator of variation of ecosystem functions [25] (Deng et al., 2019). Previous studies reported that secondary forest succession can result in remarkably shifts in the fungal communities toward that of the original fungal community in Loess Plateau [26] (Chai et al., 2019) and Lowland Forests [27] (Lin et al., 2021). Our previous study revealed that soil carbon metabolism ability obviously varied with forest types such as broadleaf forests (*Betula dahurica*), coniferous forests (*Larix gmelinii*), and conifer broadleaf forest (mixed *Larix gmelinii* and *Betula dahurica*) in Heilongjiang Zhongyangzhan Black-billed Capercaillie Nature Reserve [28] (Weng et al., 2021). Unfortunately, the fungal community structure and diversity in response to changes in soil physico-chemistry properties in these forest vegetation types have not been studied. 

The Heilongjiang Zhongyangzhan Black-billed Capercaillie Nature Reserve (established in 2013) is located in the Elhuli Mountain region of northern China. The native forest vegetation is dominated by *Larix gmelinii* as the climax community. However, over-exploitation of this resource over the past 100 years has led to drastic landscape changes. Therefore, most current forests are naturally generated secondary forests with various successional stages, such as broad-leaved conifer forests comprising of *Betula dahurica*, *Betula platyphylla*, *Larix gmelinii*, and *Quercus mongolica*, which provide an outstanding opportunity to assess the effects of vegetation on soil fungal communities under the same climate conditions. 

Here, high-throughput sequencing was conducted to characterize the ITS1 rRNA region of fungi to examine the effects of forest succession on both their structure and diversity. Specifically, we aimed to determine (i) how fungal communities respond to forest succession, and (ii) which soil properties are the main determinants of fungal diversity and community structure along with forest succession.

## 2. Material and Method

### 2.1. Site Description

The study was carried out in the mountainous areas of the Heilongjiang Zhongyangzhan Black-billed Capercaillie Nature Reserve (126°00′–126°45′ N, 48°30′–48°50′ E; Figure 1). This reserve, which belongs to the Elhuli Mountain range, covers an area of 46,743 ha. The region exhibits a cold temperate continental monsoon climate, with prolonged and cold winters and short warm wet summers, and wide fluctuations in temperature throughout the day. The mean temperature is −0.4 °C, with a maximum temperature of 37 °C and a minimum temperature of −48 °C. The average annual precipitation is 450–550 mm. The soil is deeper than 25 cm, developed from granite and classified as Borice Luvisols, according to the American Soil Taxonomy. The study site exhibits varied and unique biodiversity and therefore its preservation has garnered both national and worldwide attention. Here, we selected four naturally regenerated secondary forests and a natural climax forest (LG) as a reference. Table 1 summarizes the qualities of the five sampling sites selected herein.

### 2.2. Soil Sampling

Soil samples were acquired in July 2019 from the five selected forest types (*Betula dahurica* (BD), *Betula platyphylla* (BP), *Larix gmelinii* (LG), conifer–broadleaf forest (LGQM), and *Quercus mongolica* (QM)) (Table 1), and all the five forest types studied were located in the international monitoring site of 25 ha. Within each forest type, three sampling plots (*n* = 3) were established; dominant tree species, canopy cover, and dominant understory species were considered among the three plots to avoid unrepresentative areas. Within each plot, soil samples (0–20 cm) were collected, using an 8 cm diameter soil auger, from 15–20 locations along an S-shaped path, after removing superficial debris (e.g., leaves and dry vegetation), and mixed to ensure that the soil samples for each plot were representative. The soil samples were then transferred to sterilized press seal bags and stored in an icebox. Next, the samples were taken to the laboratory and immediately sieved (2 mm mesh) to remove stones and plant materials. One portion of the samples was air-dried to conduct physical and chemical analyses, whereas the remainder was kept at −80 °C until required for microbial analysis.

### 2.3. Characterization of Soil Physicochemical Parameters

A soil–water (1:2.5 *w*/*v*) suspension was first shaken for 30 min to measure the soil pH using a pH meter. The total N and organic C of the soil were quantified using an Elemental Analyzer (Elementar, Langenselbold, Germany). Available nitrogen (AN) was sequentially digested in H_2_SO_4_-HClO_4_, 0.5 M NaHCO_3_, and 2.0 M KCl, and then examined with a continuous flow analysis system (SKALAR SAN++, Breda, the Netherlands). After wet digestion with HClO_4_–H_2_SO_4_, total phosphorus was measured with a spectrophotometer. Available phosphorous (AP) was quantified using a colorimetric method upon extraction with 0.5 M NaHCO_3_.

### 2.4. DNA Extraction, PCR Amplification, and MiSeq Sequencing

Using the Fast DNA SPIN Extraction Kit (MP Biomedicals; Santa Ana, CA, USA), genomic DNA was extracted from 0.5 g of soil following the manufacturer’s guidelines. DNA quantity and quality were first detected by agarose gel electrophoresis (1%) and then detected by using a NanoDrop ND-1000 spectrophotometer (Thermo Fisher Scientific; Waltham, MA, USA). The fungal ITS1 region was targeted using the ITS1F (5′-CTT GGT CAT TTA GAG GAA GTA A-3′) and ITS2 (5′-GCT GCG TTC TTC ATC GAT GC-3′) primer set [29] (White et al., 1990). A 7 bp sample-specific DNA barcode was incorporated into the primers for each unique sample to allow for multiplex sequencing. All PCR procedures were conducted in two steps. First, PCR was conducted in triplicate (i.e., three independent 25 µL reactions per DNA sample). Each reaction contained 12.5 µL of 5x PCR mixed reaction buffer, 1 µL (10 µM) of forward and reverse primers, 2 µL of DNA template, and 9.5 µL of ddH2O. The PCR procedure consisted of a 10 min denaturation step at 96 °C followed by 32 cycles at 95 °C for 50 s, 55 °C for 60 s, and 72 °C for 60 s, and finally 72 °C for 10 min. The PCR products were purified using the TAKARA DNA Gel Extraction Kit (TAKARA Biosciences). Afterward, the PCR products were pooled at equal proportions and sequenced conducted by Illumina MiSeq PE300 sequencer.

### 2.5. Analysis of Sequencing Data

The raw reads were analyzed using QIIME2 [30] (Bolyen et al., 2019) and the UPARSE pipeline. Specifically, UPARSE was used for taxonomical assignment at a >97% similarity threshold [31] (Edgar, 2013). The Unite 8.0 database for fungi was used to carry out taxonomic classification. The raw sequences were deposited to in the SRA database with the accession number PRJNA691096. Operational taxonomic identities were determined using QIIME by executing the BLAST algorithm against sequences in the UNITE 8.0 database [32] (Nilsson et al., 2018). Next, the abundance of each operational taxonomic unit (OTU) per sample and the taxonomy of these OTUs were tabulated. The α-diversity indices (e.g., ACE index, Chao1 index, Simpson index, and Shannon index) were calculated in QIIME2 using the aforementioned OTU table.

### 2.6. Statistical Analyses

Venn diagrams were generated to visualize the unique and shared OTUs among the samples in R v3.3.2 [33] (R Development Core Team, 2017) using the “Venn Diagram” package. Similarly, heatmaps of the top 50 genera per sample were generated in R using the “ggplot” and “pheatmap” packages [33] (R Development Core Team, 2017). Non-metric multidimensional scaling (NMDS) based on Bray–Curtis matrices was carried out to characterize the composition of the soil fungal community using the “vegan” R package. Linear discriminant analysis (LDA) effect size (LEfSe) analysis was conducted to identify fungal indicator taxa on the basis of a normalized relative abundance matrix across groups using the default parameters. The analysis pipeline was constructed in the Galaxy platform using online interactive tools for the analysis of fungal taxonomy. One-way ANOVA was conducted to identify variations in soil physicochemical parameters, fungal α-diversity, and the relative abundance of different fungal taxa of different forest soils both at the phylum and genus level, after which least significant difference tests were conducted. All analyses were conducted using SPSS version 19.0 (Chicago, IL, USA). Non-metric multidimensional scaling based on Bray–Curtis dissimilarity was conducted in R using the “vegan” package. The correlation between fungal diversity, fungal community structure, and soil characteristics was determined via Pearson correlation analysis. The link between dominant fungal groups and soil characteristics was determined via canonical correspondence analysis in R using the “vegan” package.

## 3. Results

### 3.1. Physicochemical Characteristics of Soil

All soil physicochemical parameters measured herein differed significantly among the five forest types (Table 2). Only the soil pH in the BD forest (4.6) was significantly lower than those in other forests (5.3–5.9; *p* > 0.05) (Table 2). QM soil had the highest SOC, TN, C/N, TP, AN, and AP values, whereas LGQM had the lowest SOC and TN values. LG had the lowest C/N, TP, AN, and AP values (Table 2).

Upon conducting principal component analysis (PCA), the PC1 and PC2 represented 32.49% and 24.43% of the total variance (56.92%), respectively. Furthermore, the five forest types assessed herein exhibited a clear separation, as illustrated in the biplot (Figure 2). Taking Axis 1 as a reference, the LG forest vegetation was located in the first quadrant, whereas BP and BD were located in the second and third quadrant, respectively, and LGQM was located in the fourth quadrant (Figure 2). Additionally, there was a clear separation among the different soil parameters examined herein, as demonstrated by their distinct distribution in the PCA biplot (Figure 2). pH was located in the fourth quadrant, whereas all of the remaining parameters were situated in the second and third quadrants (Figure 2).

### 3.2. Effect of Different Vegetation Types on Fungal Diversity 

The α-diversity of the soil fungi exhibited wide variations among the five forest types examined herein (*p* < 0.05, Table 3). The BP forest soil exhibited the highest OTU number (400), Shannon index (6.30), and Simpson index (0.97), whereas BD soil had the lowest OTU number (278) and QM had the lowest Shannon index (2.84) and Simpson index (0.64) (Table 3). No significant changes in ACE index and Chao1 indices were observed among the BP, LG, LGQM, and QM soils; however, these indices were significantly lower in BD soil (Table 3). 

According to Pearson correlation analysis, the OTU number (r = 0.518, *p* < 0.05), Simpson index (r = 0.691, *p* < 0.01), and Shannon index (r = 0.806, *p* < 0.01) were positively correlated with soil pH, whereas OTU number, Chao1 index, and ACE index were positively related with SOC, C/N, and AP (Table 4). Additionally, OTU number was positively related with TP (r = 0.640, *p* < 0.01) (Table 4).

### 3.3. Effects of Different Vegetation Types on Fungal Community Structure

Upon filtering out chimeric and low-quality sequences, a total of 1,163,559 high-quality ITS1 sequences were recovered, with an average of 77,570 sequences per soil sample. Among these sequences, a total of 6113 fungal OTUs were identified at the phylum level after quality filtering. Furthermore, averages of 382 OTUs were identified in each sample. The BP forest exhibited the highest OTU number (400), whereas only 278 OTUs were identified in the BD forest (Figure 3). Rarefaction curves, diversity, and richness were obtained by randomly selecting 50,455 reads from each sample. The curve tended to flatten with sequence number at a 3% dissimilarity threshold (Figure 3), thus confirming that the obtained data were highly representative of the forest soil fungal community composition.

All the identified sequences were assigned to seven phyla and one unknown category. Basidiomycota (63.95%), Ascomycota (17.80%), Mortierellomycota (8.55%), and Mucoromycota (7.86%) were the primary phyla in the examined soil samples, accounting for more than 5% of the overall communities (Figure 4). Furthermore, despite being considerably less abundant (<5% of the fungal diversity), Glomeromycota and Chytridiomycota were identified in all of the soil samples. The different forest types had a significant effect on the relative abundances of the most abundant phyla (Appendix A). Basidiomycota, Ascomycota, and Mortierellomycota were significantly changed in five forest vegetation types (*p* > 0.05, Appendix A). Moreover, Basidiomycota was significantly more abundant in the QM forest soil compared to all other phyla, whereas Ascomycota was the least abundant phylum in the QM forest soil. In contrast, Ascomycota was the most abundant phylum in the LGQM forest soil, but Basidiomycota was least in the LGQM. Mortierellomycota was significantly more abundant in the BP forest, whereas less abundant in the QM forest (Figure 4 and Appendix A). 

The dominant genera in the soil samples were *Tricholoma* (13.51%), *Russula* (13.40%), *Cortinarius* (9.62%), *Inocybe* (7.81%), *Umbelopsis* (5.39%), *Cenococcum* (2.98%), *Boletinus* (2.97%), *Amanita* (2.20%), *Mortierella* (2.10%), *Archaeorhizomyces* (1.96%), *Clavulina* (1.79%), *Lactarius* (1.22%), and *Suillus* (1.19%) (Appendix A). According to the one-way ANOVA (Appendix A), the most five abundance genera of *Russula*, *Tricholoma*, *Cortinarius*, and *Inocybe* changed significantly in five forest vegetation types. *Tricholoma* was the most abundant genus in the QM forest soil, accounting for 58.77% of the total diversity. In contrast, *Russula* was most abundant in LGQM soil (Appendix A). 

The structures of the fungal communities, particularly the unique and shared OTUs among the samples, were visualized using Venn diagrams. A total of 194 OTUs were shared between all the forest types examined herein (Figure 5). Among the remaining OTUs, 8 were only identified in BD, 39 in BP, 39 in LG, 13 in LGQM, and 1 in QM (Figure 5).

The NMDS analysis results demonstrated that each revegetation type was accompanied by a unique fungal diversity (Figure 6). Heatmap analysis using the 30 most abundant fungal genera indicated that the genera *Tricholoma*, *Clavulina*, *Cenococcum,* and *Russula* were dominant in the QM forest (Figure 7). Furthermore, the LG forest was dominated by *Inocybe*, *Boletinus*, *Suillus*, *Penicillium*, *Discocistella*, *Sebacinaceae*, *Mortierella*, *Umbelopsis*, *Cenococcum*, and *Russula* (Figure 7). *Inocybe*, *Lactarius*, *Clavaria*, *Mortierella*, *Umbelopsis*, *Russula*, and *Cortinarius* were the most dominant genera in the BP forest. The most dominant genera in the BD forest were *Inocybe*, *Archaeorhizomyces*, *Amanita*, *Umbelopsis*, *Russula*, and *Cortinarius* (Figure 7). *Archaeorhizomyces*, *Tricholoma*, *Inocybe*, *Russula*, and *Clavulina* were dominant in the LGQM forest (Figure 7). 

### 3.4. Indicator Species

Significant differences were observed among 22 taxa in the five forest types, as indicated by LDA effect size scores of >4.5 (Figure 8A). Furthermore, 11 taxa exhibited significant differences among the forest types, with LDA > 5.0 (Figure 8B). 

### 3.5. Primary Shapers of Fungal Community Structure

The cumulative variations in the first and second RDA axes were 27.20% and 13.18, respectively (Figure 9), which demonstrated that the structure of the fungal communities was substantially influenced by the physicochemical properties of the soil.

Table 5 summarizes the five most abundant phyla and the 15 most abundant genera according to different soil physicochemical properties. At the phylum level, our findings indicated that the relative abundance of Ascomycota was negatively correlated with soil AN (r = −0.613, *p* < 0.05) and AP (r = −0.708, *p* < 0.01) (Table 5). At the genus level (Table 5), *Cortinarius* and *Amanita* abundances were negatively correlated with pH, whereas those of *Clavulina* and *Hygrocybe* were positively correlated. *Tricholoma*, *Umbelopsis,* and *Cenococcum* were positively correlated with SOC, whereas *Russula*, *Archaeorhizomyces*, and *Thelephora* were negatively correlated. *Umbelopsis* and *Cenococcum* were positively correlated with TN; in contrast, *Russula*, *Archaeorhizomyces*, and *Hygrocybe* were negatively correlated. *Tricholoma*, *Umbelopsis*, and *Clavulina* were positively correlated with C/N, whereas *Inocybe* was negatively correlated with this parameter. *Tricholoma* and *Umbelopsis* were positively correlated with TP, whereas *Inocybe*, *Boletus*, and *Suillus* were negatively correlated. *Umbelopsis* was positively correlated with AN, while *Inocybe*, *Boletus*, and *Suillus* were negatively correlated. *Tricholoma*, *Umbelopsis*, and *Clavulina* were positively correlated with AP, whereas *Inocybe*, *Boletus*, and *Suillus* were negatively correlated. 

## 4. Discussion

### 4.1. Characteristics of Soils in the Naturally Restored Forests

The soil nutrient concentrations varied substantially among the five vegetation types (Table 2). Specifically, the broadleaved forest (QM) exhibited the best soil quality, whereas the coniferous forest (LG) had the poorest soil quality, in agreement with the findings of Vitali et al. [34] (2016). These differences in soil quality were likely due to the litter characteristics and root exudates of *Quercus mongolica* [35] (Li et al., 2014). In contrast, cold, temperate, deciduous *L. gmelinii* forests exhibited higher litter amounts with higher concentrations of recalcitrant substances such as resin, lignin, wax, and tannin. Coverage of the forest floor by dense coniferous litter is thought to obstruct air circulation, thus preventing nutrient accumulation. Conversely, *Q. mongolica* commonly occurs in warm evergreen broadleaf forests and produces litter that is easily decomposed, thus enhancing nutrient quality [11] (Cheng, 2013). 

The pH of the soil in the study site ranged from 4.6 to 5.9, among which the lowest pH values were observed in the soil surrounding *B. dahurica*. These acidic conditions are likely due to litter quality. *B. dahurica* produces leaf litter with low nitrogen content, higher lignin content, high C/N ratio, and higher lignin/N relative to other types of broadleaf forests [36] (Gao et al., 2015). Similarly, variations in the characteristics of leaf litter produced by different tree species have been linked to differences in soil quality even under equal climate conditions [37] (Ge et al., 2012). These observations aligned with our findings, as the soils studied herein could be classified into five groups on the basis of their physicochemical properties (Figure 2). These differences in soil characteristics, particularly those between broadleaf and coniferous forests, may contribute greatly to shaping the community structure of soil microbes.

### 4.2. Effects of Vegetation Types on the Diversity and Structure of Fungal Communities

Our findings indicated that different forest vegetation types had unique effects on fungal community diversity and structure (Table 3; Figure 4 and Appendix A). The BP forest exhibited the highest fungal Shannon index (Table 3), which indicated that this forest type possessed the highest soil fungal diversity and evenness. Furthermore, the LG forest exhibited the highest fungal ACE index, indicating a higher fungal abundance in the soil. These findings were ascribed to the variations in the chemical composition and decomposition rate of the litter produced by the aforementioned forest types [38] (Anderson, 2009), which affect the physicochemical properties of soil and, in turn, modify the fungal diversity indices of the soil. Our observations confirmed that the effects of the tree species on fungal diversity were most likely due to the litter quantity and quality produced by the trees following revegetation. Interestingly, the different revegetation types did not affect the overall fungal community composition of the soil but did modify the relative abundances of different taxa. These outcomes were likely attributable to variations in the root residues and secretions produced by different species of trees [39] (Degrune et al., 2015). Despite the small sample sizes of our study (*n* = 3), we identified significant differences in the structure of the fungal communities associated with the different revegetation types studied herein. The fungal communities were largely dominated by the Basidiomycota phylum, followed by Ascomycota, Mortierellomycota, and Mucoromycota (Figure 4). Importantly, these findings were in line with those of Gorfer et al. [40] (2013) and Waring et al. [41] (2016). Basidiomycota typically inhabit dry and cool habitats [42] (Sui et al., 2021), and their relative abundance in soils is thought to be linked to their capacity to metabolize lignocellulose [43] (Alfaro et al., 2016), which in turn is determined by soil organic carbon dynamics [44] (Blackwood et al., 2010). Moreover, the BP forest exhibited a high relative abundance of Ascomycota, which was consistent with the findings of Gao et al. [45]. Previous studies in tropical regions have also demonstrated that broadleaf forests are largely dominated by the Ascomycota phylum [25]. The high abundance of Ascomycota in BP observed in our study suggested the presence of saprotrophs, which provide Ascomycota with easily degradable nutrient sources such as organic matter [46]. The variations in the fungal abundances in soils with different vegetation may also be due to environmental filtering and niche differentiation [47].

The *Tricholoma*, *Russula*, and *Cortinarius* genera were dominant in the soil samples examined herein (Appendix A), which was consistent with other studies [48,49]. *Russula* was among the most common genera in our study, and past literature has suggested that some members of this genus might provide water and nutrients to their host plants, thus allowing them to overcome biotic and abiotic stress [50]. The members of the genus *Cortinarius* are ubiquitous fungi that are distributed worldwide [51]. Interestingly, this genus was also among the most common in our study. The effects of forest types on the diversity and abundance of fungi have been linked to the quantity and characteristics of the organic matter derived from plant litter, as it constitutes an important nutrient source for soil microorganism growth [52]. The heatmap in Figure 6 and the NMDS plots in Figure 7 demonstrate that variations in revegetation type shaped the fungal communities in the surrounding soil. This distinction was particularly noticeable when comparing the fungal community structures associated with broad-leaved and coniferous forests. A recent study also demonstrated that the soil fungal community composition in *Pinus koraiensis* forests was different from that in a broadleaved *Pinus koraiensis* mixed forest [25]. Therefore, our findings demonstrated the strong influence of forest and vegetation types on the structure and diversity of soil fungal communities [53].

### 4.3. Effects of Soil Characteristics on Fungal Communities

Soil properties strongly influence the structure and diversity of fungal communities. Here, OTU number, Shannon index, and Simpson index exhibited a positive correlation with pH (Table 4), which was consistent with the findings of Hartmann et al. [54] and Li et al. [55]. Moreover, OTU number, Chao1 index, and ACE index were positively correlated with available P (Table 4), which largely mirrored the findings of Liu et al. [56]. Other studies have demonstrated that several physicochemical characteristics of soil including moisture [57], pH [58], organic carbon [59], and C/N ratio [60] substantially affect the composition of fungal communities. Our findings also confirmed that the abundances of fungi at the phylum and genus levels were significantly correlated with soil pH. Furthermore, the composition of the fungal communities was significantly correlated with total N, SOC, available N, and available P (Table 5; Figure 9), which was in line with previous findings [7,53]. 

The members of the Basidiomycota phylum are highly sensitive to variations in soil physicochemical properties [42]. Here, the relative abundance of Basidiomycota was negatively correlated with AP and pH, which contrasted with previous reports [61,62]. Wang et al. [63] reported higher Ascomycota abundances in soils with higher pH values. In contrast, our study did not identify any correlation between Ascomycota abundance and soil pH, which may be due to the relatively narrow soil pH range (4.6–5.9) in our study, which translated to low variations in the abundance of Ascomycota in our study. Furthermore, the relative abundance of both Ascomycota and Mortierellomycota was negatively correlated with available P, suggesting that this parameter plays a crucial role in shaping the fungal community structure, which aligns with the reports of Dang et al. [64]. Furthermore, Wu et al. [65] recently reported that the abundance of Ascomycota was positively correlated with soil organic matter content. Taken together, our findings indicated that the composition and diversity of soil fungal communities are strongly shaped by variations in revegetation types, as different plant species can have unique effects on soil physicochemical properties.

The dominant fungal genera (*Russula*, *Tricholoma*, *Cortinarius*, and *Inocybe*) were representative of the dominant genera found in our study (Appendix A), which was accordant with previous research (Lin et al., 2021). *Russula* was the most common genus in our study, and previous studies have put forward that *Russula* was symbionts of trees and are therefore abundant in forests [54]. The influence of different forest vegetation on the soil fungal community is often related to the quantity of organic matter and nitrogen returned by plant litter, which provides major nutrition for soil fungi [25]. On the basis of previous research, *Tricholoma* has been reported to be distributed throughout the forests [66], which was also the common genus in our study. *Cortinarius* and *Inocybe* were mainly ectomycorrhizal fungi in the forests that can help its host plant to overcome biotic and abiotic stresses by supplying it with water and nutrients [67]. 

The results of clear differentiation in the NMDS (Figure 6) and heatmap (Figure 7) and Cladogram (Figure 8) plots illustrated that distinct differences in the fungal communities were observed in different forest types, suggesting that broad-leaved forests, coniferous forests, and mixed conifer–broadleaf forests each owned different fungal communities. Our results were agreement with a previous study that investigated the composition of soil fungal community in different forest types in the Korean pine plantation [25]. These findings confirmed that composition of forests with different tree species altered the soil fungal community composition and diversity. Moreover, this proves the conclusion that the different forest vegetation types shape the soil fungal community through affecting the soil physicochemistry properties [26]. 

This study investigated the fungal structure and diversity of different forest types in Zhongyangzhan Black-billed Capercaillie Nature Reserve. This study provided data support for understanding the changes of soil microorganisms during the succession of high-latitude forest vegetation, and also enriched the impact of forest vegetation changes on the variation of microorganisms in soil. In high-latitude, cold-region forest ecosystems, in the process of forest succession, in addition to being affected by soil physicochemical properties and vegetation types, the litter composition, root biomass, and exudates all affect the composition and composition of soil fungi. Therefore, multi-dimensional comprehensive experiments will be carried out in the future to explain the response mechanism of soil microbial community structure and function to environmental heterogeneity during forest succession.

## 5. Conclusions

Our findings indicated that the fungal community of the Heilongjiang Zhongyangzhan Black-billed Capercaillie Nature Reserve is largely dominated by the Basidiomycota, Ascomycota, Mortierellomycota, and Mucoromycota phyla, and their relative abundances varied significantly in a revegetation type-dependent manner. Moreover, the different revegetation types significantly affected the OTU number and the ACE, Shannon, Simpson, and Chao1 indices of the fungal communities of the study site. Furthermore, the abundances of the most dominant phyla and genera were significantly correlated with soil pH, SOC, total N, available P, total P, and available N. Collectively, our results demonstrated the effects of vegetation type on soil fungal communities could be attributed to the quantity and qualities of the litter derived from different vegetation types, as this can lead to variations in soil quality even in regions with equal climate conditions. This was particularly noticeable when comparing the fungal community composition and diversity of broad-leaved and coniferous forests.

## Figures and Tables

**Figure 1 microorganisms-10-00829-f001:**
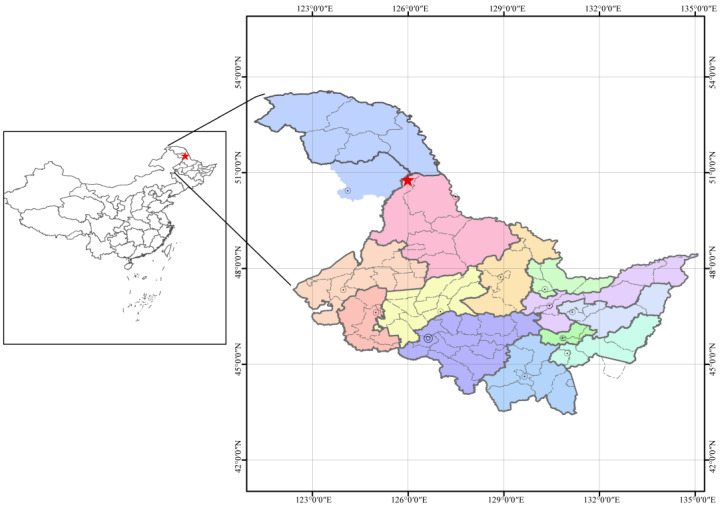
The asterisk indicates the study site in Heilongjiang Province and China.

**Figure 2 microorganisms-10-00829-f002:**
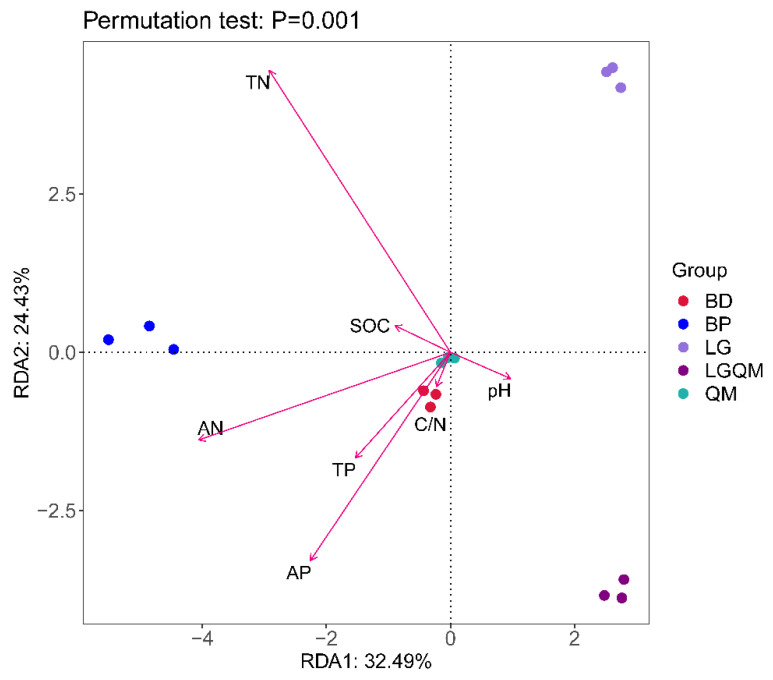
RDA results, including all of the examined soil physicochemical parameters. BD, *Betula dahurica* forest; BP, *Betula platyphylla* forest; LG, *Larix gmelinii* forest; LGQM, *Q. mongolica* and *L. gmelinii* mixed forest; QM, *Quercus mongolica* forest. TN, total nitrogen; SOC, soil organic carbon; TP, total phosphorous; AN, available nitrogen; AP, available phosphorous.

**Figure 3 microorganisms-10-00829-f003:**
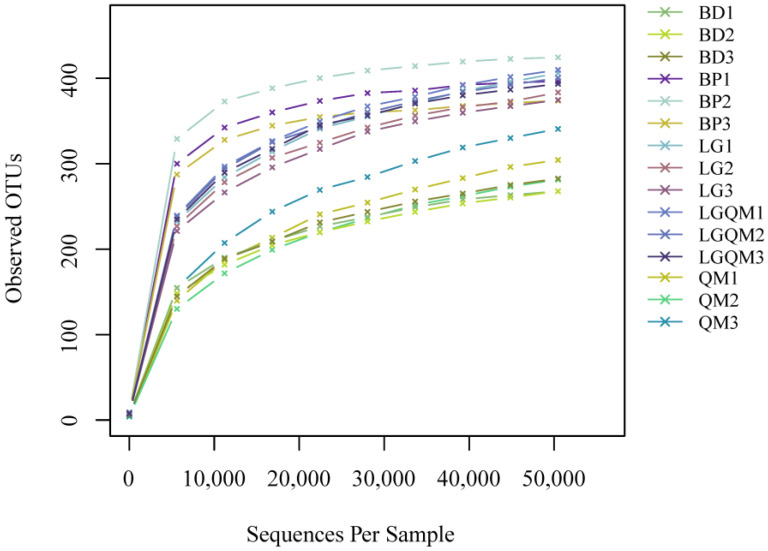
Rarefaction curves. BD1-BD3, *B. dahurica*; BP1-BP3, *B. platyphylla*; LG1-LG3, *L. gmelinii*; LGQM1-LGQM3, *Q. mongolica* and *L. gmelinii mixed* forest; QM1-QM3, *Q. mongolica*.

**Figure 4 microorganisms-10-00829-f004:**
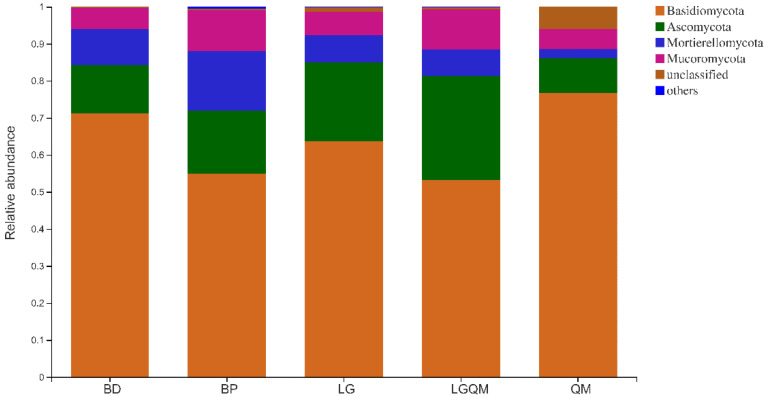
Effect of revegetation type on the relative abundance of fungal phyla. BD, *Betula dahurica*; BP, *Betula platyphylla*; LG, *Larix gmelinii*; LGQM, *Q. mongolica* and *L. gmelinii* mixed forest; QM, *Quercus mongolica*.

**Figure 5 microorganisms-10-00829-f005:**
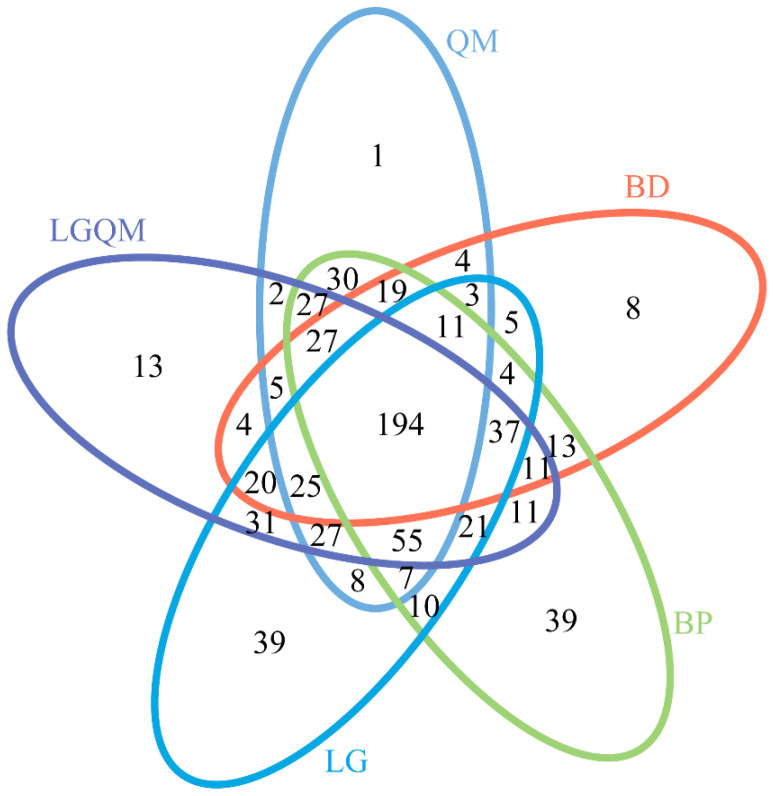
Venn diagram of shared and unique fungal OTUs among the vegetation types. BD, *Betula dahurica*; BP, *Betula platyphylla*; LG, *Larix gmelinii*; LGQM, *Q. mongolica* and *L. gmelinii* mixed forest; QM, *Quercus mongolica*.

**Figure 6 microorganisms-10-00829-f006:**
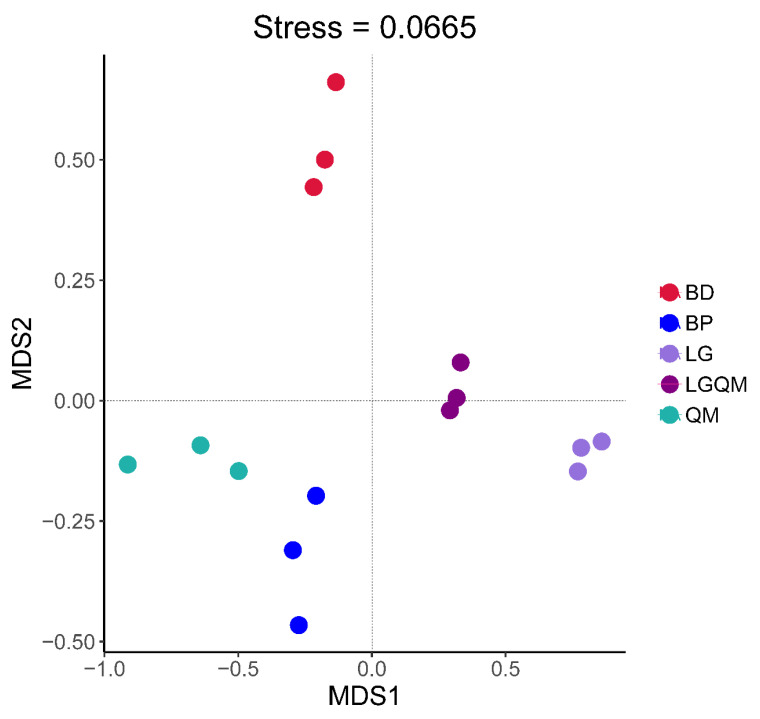
NMDS analysis of the fungal communities in the different forest type soils. BD, *Betula dahurica*; BP, *Betula platyphylla*; LG, *Larix gmelinii*; LGQM, *Q. mongolica* and *L. gmelinii* mixed forest; QM, *Quercus mongolica*.

**Figure 7 microorganisms-10-00829-f007:**
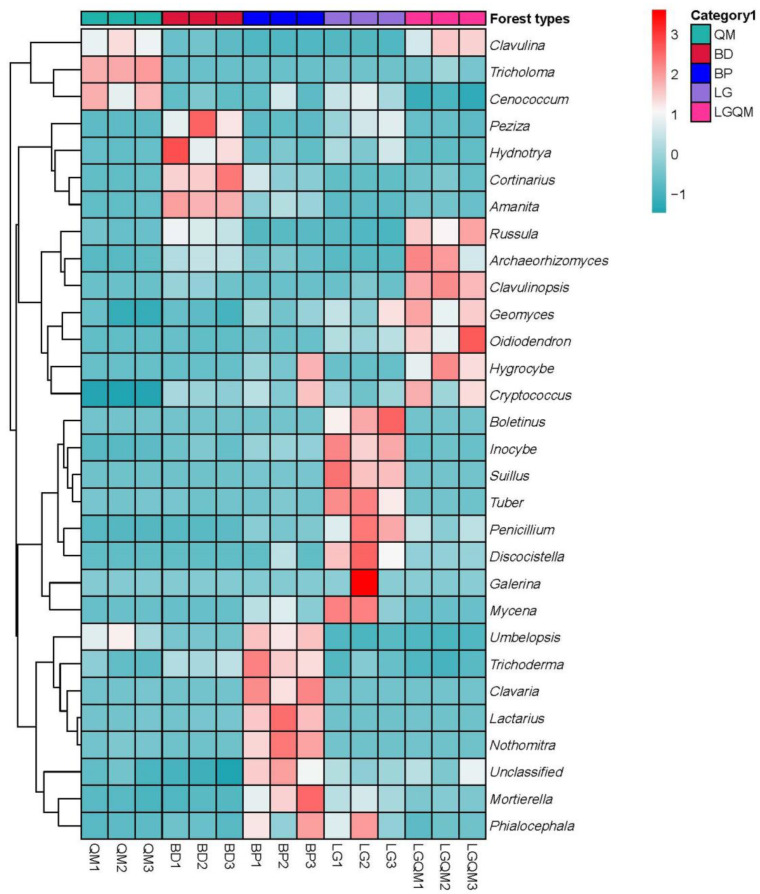
Heatmap and hierarchical clustering of the relative abundances of the top 30 genera detected in the soil fungal communities. The color gradient (red, white, blue) represents the relative abundance of the soil fungi from high to low in the different forest types. BD1–BD3, *Betula dahurica*; BP1–BP3, *Betula platyphylla*; LG1–LG3, *Larix gmelinii*; LGQM1–LGQM3, *Q. mongolica* and *L. gmelinii* mixed forest; QM1–QM3, *Quercus mongolica*.

**Figure 8 microorganisms-10-00829-f008:**
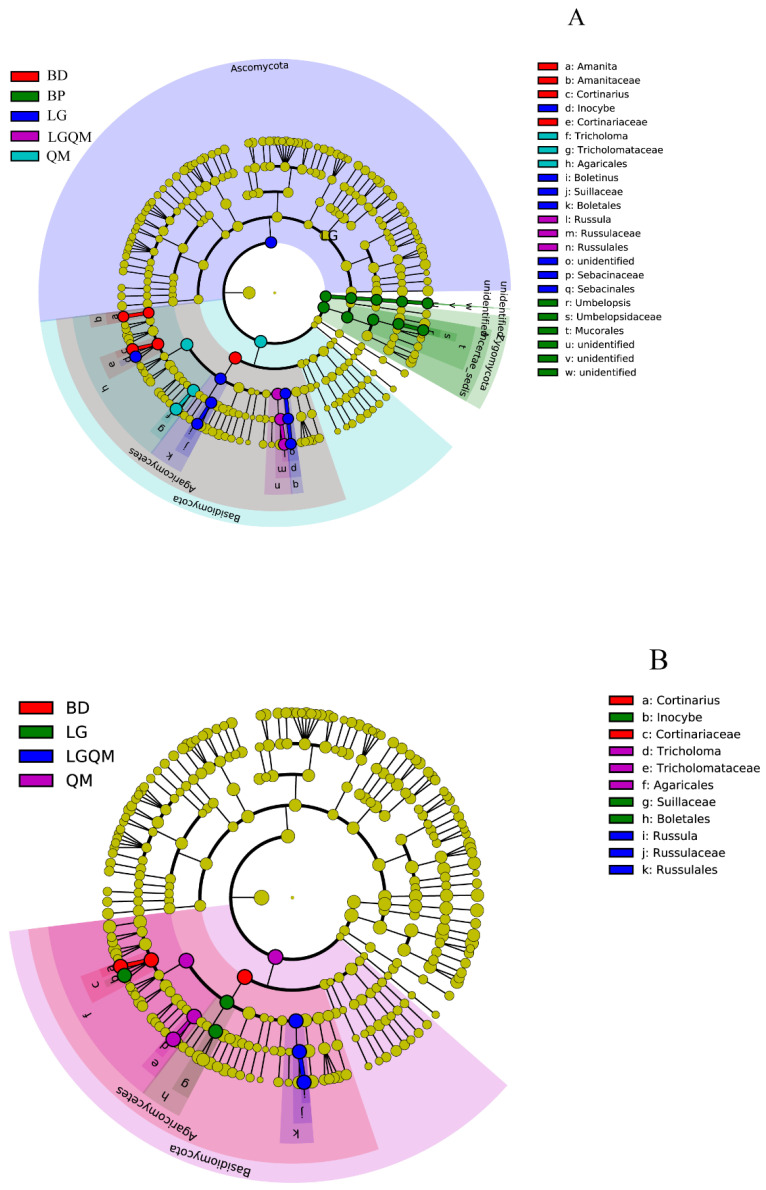
Cladogram of the fungal communities in the five types of forest soils with LDA > 4.5 (**A**) and >5.0 (**B**). The circles represent fungal taxa from phylum to genus starting from the center. BD, *Betula dahurica*; BP, *Betula platyphylla*; LG, *Larix gmelinii*; LGQM, *Q. mongolica* and *L. gmelinii* mixed forest; QM, *Quercus mongolica*.

**Figure 9 microorganisms-10-00829-f009:**
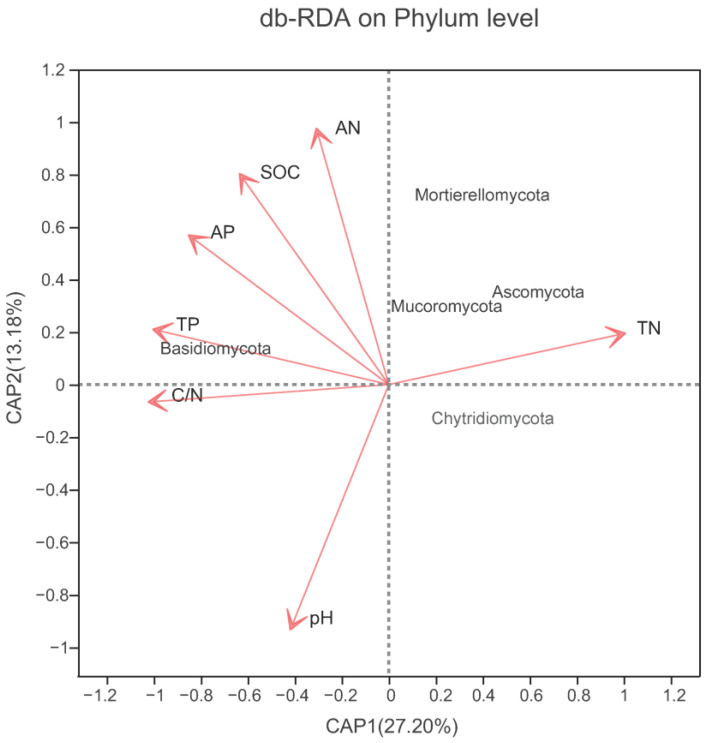
RDA of the dominant fungal phyla in soil constrained by soil variables.

**Table 1 microorganisms-10-00829-t001:** Characteristics of the five naturally restored forest types in the Heilongjiang Zhongyangzhan Black-billed Capercaillie Nature Reserve.

Vegetation Type	Primary Tree Species	Elevation (m)	Forest Type
BD	*Populus tremula*, *Betula dahurica*	552.1	Natural secondary forest
BP	*Betula platyphylla*, *Populus tremula*	535.5	Natural secondary forest
QM	*Quercus mongolica*, *Tilia amurensis*	520.4	Natural secondary forest
LGQM	*Betula platyphylla*, *Populus tremula*, *Quercus mongolica*, *Larix gmelinii*	565.4	Natural secondary forest
LG	*Larix gmelinii*	585.7	Natural climax forest

Notes: BD, *Betula dahurica*; BP, *Betula platyphylla*; LG, *Larix gmelinii*; LGQM, *Q. mongolica* and *L. gmelinii* mixed forest; QM, *Quercus mongolica*.

**Table 2 microorganisms-10-00829-t002:** Physicochemical characteristics of the soil samples in the five restored forest types.

Soil Physicochemical Parameters	LG	BD	LGQM	BP	QM	F-Value	*p*-Value
pH	5.5 ± 0.1 a	4.6 ± 0.4 b	5.9 ± 0.3 a	5.3 ± 0.1 a	5.4 ± 0.2 a	10.233	*p* < 0.001
SOC (g/kg)	60.3 ± 3.5 c	54.0 ± 3.53 d	49.9 ± 3.95 d	76.2 ± 2.04 b	107.4 ± 22.4 a	191.954	*p* < 0.001
TN (g/kg)	4.1 ± 0.1 a	3.47 ± 0.34 b	2.76 ± 0.13 c	4.27 ± 0.15 a	4.20 ± 0.10 a	33.856	*p* < 0.001
C/N	14.7 ± 1.3 c	15.59 ± 0.76 c	18.07 ± 1.58 b	17.86 ± 0.35 b	25.82 ± 0.61 a	55.253	*p* < 0.001
TP (g/kg)	1.77 ± 0.15 c	2.07 ± 0.06 bc	2.17 ± 0.29 bc	2.33 ± 0.15 ab	2.60 ± 0.20 a	8.374	*p* < 0.001
AN (mg/kg)	24.63 ± 2.60 d	78.67 ± 2.96 b	38.59 ± 2.18 c	83.75 ± 2.22 a	87.34 ± 1.59 a	450.879	*p* < 0.001
AP (mg/kg)	27.43 ± 0.81 c	36.17 ± 1.23 b	35.86 ± 1.03 b	37.28 ± 0.42 b	41.25 ± 1.40 a	88.517	*p* < 0.001

Note: The results in the table include the F- and *p*-values from ANOVA. Different letters in each row indicate significant differences (*p* < 0.05) among the five reforestation types tested herein. BD, *Betula dahurica* forest; BP, *Betula platyphylla* forest; LG, *Larix gmelinii* forest; LGQM, *Q. mongolica* and *L. gmelinii* mixed forest; QM, *Quercus mongolica* forest; SOC, soil organic carbon; TN, total nitrogen; TP, total phosphorus; AN, available nitrogen; AP, available phosphorous.

**Table 3 microorganisms-10-00829-t003:** Alpha diversity of the soil fungi in the forest types examined herein.

Types	Richness	Shannon Index	ACE Index	Chao1 Index	Simpson Index
BD	*278 ±**9* c	4.21 ± 0.31 c	*326.57 ±**16.90* b	*333.84 ±**29.21* b	0.86 ± 0.04 b
BP	**400 ± 25** a	**6.30 ± 0.18** a	410.24 ± 23.94 a	419.49 ± 20.88 a	**0.97 ± 0.00** a
LG	395 ± 17 a	5.28 ± 0.14 b	**453.18 ± 29.73** a	447.63 ± 28.58 a	0.94 ± 0.01 a
LGQM	408 ± 9 a	5.36 ± 0.04 b	450.89 ± 15.21 a	**451.24 ± 18.81** a	0.94 ± 0.00 a
QM	318 ± 30 b	*2.81 ±**0.07* d	396.44 ± 36.84 a	401.18 ± 43.13 a	*0.64 ±**0.03* c

Notes: Significant differences (*p* < 0.05) are indicated by the different letters in each column. The abbreviations are the same as those in Table 2. The highest and lowest values in each column are indicated in bold and italics, respectively.

**Table 4 microorganisms-10-00829-t004:** Person’s rank correlation coefficients between fungal alpha-diversity and soil parameters.

	pH	SOC	TN	C/N	TP	AN	AP
Richness	0.52 *	0.54 *	0.12	0.79 **	0.53 *	0.20	0.64 *
ACE	0.21	0.58 *	0.04	0.75 **	0.45	0.37	0.66 **
Chao1	0.24	0.61 *	0.03	0.78 **	0.50	0.37	0.67 **
Shannon	0.81 **	0.06	0.48	0.40	0.27	0.29	0.24
Simpson	0.70 **	−0.21	0.71 **	0.21	−0.15	−0.36	0.16

Notes: * 0.05 (one-tailed); ** 0.01 (two-tailed).

**Table 5 microorganisms-10-00829-t005:** Person’s rank correlations between the relative abundances of dominant bacteria taxa and soil physicochemical parameters.

Fungal Taxa	pH	SOC	TN	C/N	TP	AN	AP
**Phylum**	-	-	-	-	-	-	-
Basidiomycota	−0.294	0.230	−0.024	0.283	0.097	0.205	0.192
Ascomycota	0.474	−0.487	−0.178	−0.480	−0.389	−0.708 **	−0.613 *
Mortierellomycota	−0.210	−0.134	0.332	−0.378	−0.121	0.228	−0.056
Mucoromycota	0.170	−0.026	0.093	−0.068	0.099	0.072	0.066
Glomeromycota	0.152	−0.016	0.119	−0.090	0.210	0.253	0.213
**Genus**	-	-	-	-	-	-	-
*Tricholoma*	0.220	0.827 **	0.279	0.880 **	0.625 *	0.411	0.607 *
*Russula*	0.036	−0.564 *	−0.948 **	−0.116	−0.001	−0.180	0.145
*Cortinarius*	−0.727 **	−0.351	−0.152	−0.385	−0.127	0.402	0.128
*Inocybe*	0.100	−0.276	0.344	−0.565 *	−0.669 **	−0.676 **	−0.904 **
*Umbelopsis*	−0.164	0.707 **	0.617 *	0.515 *	0.648 **	0.763 **	0.607 *
*Cenococcum*	0.025	0.722 **	0.669 **	0.501	0.295	0.235	0.141
*Boletus*	0.193	−0.194	0.266	−0.412	−0.689 **	−0.698 **	−0.844 **
*Amanita*	−0.804 **	−0.363	−0.193	−0.374	−0.137	0.427	0.134
*Mortierella*	0.130	−0.030	0.433	−0.297	−0.147	−0.048	−0.250
*Archaeorhizomyces*	0.203	−0.609 *	−0.869 **	−0.239	−0.187	−0.265	0.092
*Clavulina*	0.518 *	0.264	−0.479	0.651 **	0.499	0.006	0.520 *
*Lactarius*	−0.109	0.161	0.425	−0.056	0.230	0.440	0.213
*Suillus*	0.161	−0.222	0.313	−0.475	−0.636 *	−0.725 **	−0.890 **
*Thelephora*	0.332	−0.524 *	−0.458	−0.355	−0.045	−0.249	−0.096
*Hygrocybe*	0.552 *	−0.329	−0.520 *	−0.079	0.015	−0.232	0.095

Notes: * 0.05 (one-tailed); ** 0.01 (one-tailed).

## Data Availability

Raw data of fungal sequences were deposited into the NCBI Sequence Read Archive under acces-sion number PRJNA691096.

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
