# Peer review of "Influence of Different Vegetation Types on Soil Physicochemical Parameters and Fungal Communities"

_microorganisms, 2022, doi:10.3390/microorganisms10040829_

Round 1
Reviewer 1 Report
This study investigated the fungal communities in the different vegetation types. The fungal communities were analyzed using a metabarcoding approach, and showed a significant difference between vegetation types. Given the importance of fungi in an ecosystem, the subject is interesting and deserves publication. However, several points need to be revised well before the publication.
L128-129: Please add the citation for the original reference of primers.
L141: QIIME v1.9.0 is not QIIME2, but QIIME1. Please check and correct it.
L143: Change Unite to UNITE and add the citation.
Table 2: check the bold letter in BP.
Figure 4: This plot is wrong according to Figure S2. There are too much of unclassified fungi. For example, unclassified_f__Sebacinaceae is not “unclassified” but the Basidiomycota. Please check and correct it.
Author Response
Dear Reviewer,
Thank you for your letter and for the comments concerning our manuscript entitled “Influence of different vegetation types on soil physicochemical parameters and fungal communities (No. 1625020)”. Those comments are all valuable and very helpful for revising and improving our paper. We have studied all provided comments carefully and have made appropriate corrections which we hope meet with approval. The corrections made in the paper and the respective responses to your comments are listed below and shown by highlight yellow words in the improved version of the text.

Reviewer 2 Report
Peer-review of Manuscript microorganisms-1625020
Line 21: Why “Capercaillie” is italics? Elsewhere not. Make this uniform.
Line 67: “…few efforts have been made to study the effects of forest restoration…” The claim that there are few studies on the litter communities of planted/restored forests is a hazardous one. A quick search of the literature reveals many studies, including recent ones. I suggest that the authors implement the introduction with a more thorough, wide-ranging, and also more comprehensive literature search. As an example, some important papers on the topic not quoted or commented could be the following: 10.3389/fmicb.2021.676251 and 10.3389/fmicb.2019.00895 or 10.1128/aem.00966-17
Line 79: the question “(i) how fungal communities respond to different regenerated vegetation types” seems badly posed. It is unclear, also beacuse using the term 'type', when perhaps "vegetation association" or another more appropriate term would help. Is reference made to newly planted forests? replanted with the same species? naturally modified according to a secondary succession? It is necessary to explain more the purpose of the work using appropriate terminology.
Line 89: give max and min temperatures, not only the mean temperature
Line 90: Furthermore, the region is characterized by dark brown soil (Zhang et al., 2017): In a work in which the focus is on soil, it is unacceptable that the soil classification of the study area is not reported in detail. A simple note "dark brown soil" has no scientific significance. Add pedological information on the sampling sites. In addition to this serious inaccuracy, there is also the type of sampling carried out, which shows how the authors, despite having studied forest soils, did not consider the structure and pedogenesis of these soils, made up of horizons and profiles with a precise classification. The analysis carried out is relative to a fixed volume of soil (0-20 cm), which makes the entire study rather useless, since the data are probably relative to a set of micro-environments (organic and mineral horizons, etc.) taken together and mixed.
What justification do the authors have? Why such a sampling that disrespects the nature and formation of the soils studied? How could they have compared equal volumes but probably different bulk densities? Soil bulk density: a measurement that has unfortunately not been reported…
Any differences in the chemical composition of the different sites could be the result of initial sampling error or mixing horizons with different characteristics.
The whole fungal community analysis is conducted in a standard and rather tedious and pointless way (it is the same in all the outputs of some well-known sequencing companies) and adds very little to the discussion that could have been had with more ecological value.
Lines 421-22 The authors conclude that the study area is largely dominated by the Basidiomycota, Ascomycota, Mortierellomycota, and Mucoromycota phyla. This is a rather trivial statement that does not take into account the fact that the largest percentage of the sequences at all sites were not assigned (unknown category). See also in the results (Line 233) where it is stated that "Basidiomycota, Ascomycota, Mortierellomycota, and Mucoromycota were the primary phyla in the examined soil samples, accounting for more than 1% of the soil fungal diversity" One percent seems very little to indicate primary phyla, is there a mistake perhaps? How do the authors comment on this unintuitive statement? Any chance to go deeper with the analysis of the sequences?
Author Response

(The authors gave the same response as above.)

Round 2
Reviewer 1 Report
The authors have revised the manuscript well as following the comments. It can be recommended for publication after minor revision.
L160: Add the “v” before the 8.0 to indicate version information (v. 8.0).
Table 2: Please use italics for the scientific names (e.g. Betula dahurica).